# Learning to Set Waypoints for Audio-Visual Navigation

**Changan Chen**[1,2]   **Sagnik Majumder**[1]   **Ziad Al-Halah**[1]   **Ruohan Gao**[1,2]
**Santhosh K. Ramakrishnan**[1,2]   **Kristen Grauman**[1,2]
[1]UT Austin   [2]Facebook AI Research

## Abstract

In audio-visual navigation, an agent intelligently travels through a complex, un-mapped 3D environment using both sights and sounds to find a sound source (e.g., a phone ringing in another room). Existing models learn to act at a fixed granularity of agent motion and rely on simple recurrent aggregations of the audio observations. We introduce a reinforcement learning approach to audio-visual navigation with two key novel elements: 1) waypoints that are dynamically set and learned end-to-end within the navigation policy, and 2) an acoustic memory that provides a structured, spatially grounded record of what the agent has heard as it moves. Both new ideas capitalize on the synergy of audio and visual data for revealing the geometry of an unmapped space. We demonstrate our approach on two challenging datasets of real-world 3D scenes, Replica and Matterport3D. Our model improves the state of the art by a substantial margin, and our experiments reveal that learning the links between sights, sounds, and space is essential for audio-visual navigation. Project: `http://vision.cs.utexas.edu/projects/audio_visual_waypoints`.

## 1   Introduction

Intelligent robots must be able to move around efficiently in the physical world. In addition to geometric maps and planning, work in embodied AI shows the promise of agents that *learn* to map and navigate. Sensing directly from egocentric images, they jointly learn a spatial memory and navigation policy in order to quickly reach target locations in novel, unmapped 3D environments (Gupta et al., 2017b;a; Savinov et al., 2018; Mishkin et al., 2019). High quality simulators have accelerated this research direction to the point where policies learned in simulation can (in some cases) successfully translate to robotic agents deployed in the real world (Gupta et al., 2017a; Müller et al., 2018; Chaplot et al., 2020b; Stein et al., 2018).

Much current work centers around visual navigation by a PointGoal agent that has been told where to find the target (Gupta et al., 2017a; Sax et al., 2018; Mishkin et al., 2019; Savva et al., 2019; Chaplot et al., 2020b). However, in the recently introduced AudioGoal task, the agent must use both visual and auditory sensing to travel through an unmapped 3D environment to find a sound-emitting object, without being told where it is (Chen et al., 2020; Gan et al., 2020). As a learning problem, AudioGoal not only has strong motivation from cognitive and neuroscience (Gougoux et al., 2005; Lessard et al., 1998), it also has compelling real-world significance: a phone is ringing somewhere upstairs; a person is calling for help from another room; a dog is scratching at the door to go out.

What role should audio-visual inputs play in learning to navigate? There are two existing strategies. One employs deep reinforcement learning to learn a navigation policy that generates step-by-step actions (TurnRight, MoveForward, etc.) based on both modalities (Chen et al., 2020). This has the advantage of unifying the sensing modalities, but can be inefficient when learning to make long sequences of individual local actions. The alternative approach separates the modalities—treating the audio stream as a beacon that signals the goal location, then planning a path to that location using a visual mapper (Gan et al., 2020). This strategy has the advantage of modularity, but the disadvantage of restricting audio's role to localizing the target. Furthermore, both existing methods make strong assumptions about the granularity at which actions should be predicted, either myopically for each step (0.5 to 1 m) (Chen et al., 2020) or globally for the final goal location (Gan et al., 2020).

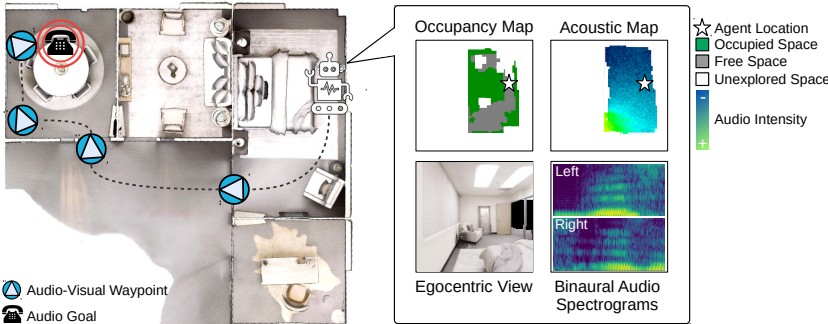

Figure 1: Waypoints for audio-visual navigation: Given egocentric audio-visual sensor inputs (depth and binaural sound), the proposed agent builds up both geometric and acoustic maps (top right) as it moves in the unmapped environment. The agent learns encodings for the multi-modal inputs together with a modular navigation policy to find the sounding goal (e.g., phone ringing in top left corner room) via a series of dynamically generated audio-visual waypoints. For example, the agent in the bedroom may hear the phone ringing, identify that it is in another room, and decide to first exit the bedroom. It may then narrow down the phone location to the dining room, decide to enter it, and subsequently find it. Whereas existing hierarchical navigation methods rely on heuristics to determine subgoals, our model learns a policy to set waypoints jointly with the navigation task.

We introduce a new approach for AudioGoal navigation where the agent instead predicts non-myopic actions with self-adaptive granularity. Our key insight is to *learn to set audio-visual waypoints*: the agent dynamically sets intermediate goal locations based on its audio-visual observations and partial map—and does so in an end-to-end manner with learning the navigation task. Intuitively, it is often hard to directly localize a distant sound source from afar, but it can be easier to identify the general direction (and hence navigable path) along which one could move closer to that source. See Figure 1.

Both the audio and visual modalities are critical to identifying waypoints in an unmapped environment. Audio input suggests the general goal direction; visual input reveals intermediate obstacles and free spaces; and their interplay indicates how the geometry of the 3D environment is warping the sounds received by the agent, such that it can learn to trace back to the hidden goal. In contrast, subgoals selected using only visual input are limited to mapped locations or clear line-of-sight paths.

To realize our idea, our first contribution is a novel deep reinforcement learning approach for AudioGoal navigation with audio-visual waypoints. The model is hierarchical, with an outer policy that generates waypoints and an inner module that plans to reach each waypoint. Hierarchical policies for 3D navigation are not new, e.g., (Chaplot et al., 2020b; Stein et al., 2018; Bansal et al., 2019; Caley et al., 2016). However, whereas existing visual navigation methods employ heuristics to define subgoals, the proposed agent *learns to set useful subgoals in an end-to-end fashion for the navigation task.* This is a new idea for 3D visual navigation subgoals in general, not specific to audio goals (cf. Sec. 2). As a second technical contribution, we introduce an *acoustic memory* to record what the agent hears as it moves, complementing its visual spatial memory. Whereas existing models aggregate audio evidence purely based on an unstructured memory (GRU), our proposed acoustic map is structured, interpretable, and integrates audio observations throughout the reinforcement learning pipeline.

We demonstrate our approach on the complex 3D environments of Replica and Matterport3D using SoundSpaces audio (Chen et al., 2020). It outperforms the state of the art for AudioGoal navigation by a substantial margin (8 to 49 points in SPL on heard sounds), and generalizes much better to the challenging cases of unheard sounds, noisy audio, and distractor sounds. Our results show learning to set waypoints in an end-to-end fashion outperforms current subgoal approaches, while the proposed acoustic memory helps the agent set goals more intelligently.

## 2 RELATED WORK

**Learning to navigate in 3D environments**  Robots can navigate complex real-world environments by mapping the space with 3D reconstruction algorithms (i.e., SfM) and then planning their movements (Thrun, 2002; Fuentes-Pacheco et al., 2012). While many important advances follow this line of work, ongoing work also shows the promise of *learning* map encodings and navigation policies directly from egocentric RGB-(D) observations (Gupta et al., 2017a;b; Savinov et al., 2018; Mishkin et al., 2019). Current methods focus on the so-called PointGoal task: the agent is given a 2D displacement vector pointing to the goal location and must navigate through free space to get there.

The agent relies on visual input and (typically) GPS odometry (Gupta et al., 2017a; Mishkin et al., 2019; Savva et al., 2019; Sax et al., 2018; Chaplot et al., 2020b).

In contrast, the recently introduced AudioGoal task requires the agent to navigate to a *sound source* goal using vision and audio (Chen et al., 2020; Gan et al., 2020). Importantly, unlike PointGoal, AudioGoal does not provide a displacement vector indicating the goal. Existing AudioGoal methods either learn a policy to select the best immediate next action using the multi-modal inputs (Chen et al., 2020), or predict the final goal location from the audio input and then follow a planned path to it based on visual inputs (Gan et al., 2020). Our ideas for audio-visual waypoints and an acoustic map are entirely novel, and lead to a significant improvement in performance.

**Navigation with intermediate goals**  Current methods often learn policies that reward moving to the final goal location using a step-by-step action space (e.g., TurnRight, MoveForward, Stop) (Gupta et al., 2017a; Mirowski et al., 2016; Mishkin et al., 2019; Savva et al., 2019). However, recent work explores ways to incorporate subgoals or waypoints for PointGoal navigation. Taking inspiration from hierarchical learning (Bacon et al., 2017; Nachum et al., 2018), the general idea is to select a subgoal, use planning (or a local policy) to navigate to the current subgoal, and repeat (Stein et al., 2018; Bansal et al., 2019; Chaplot et al., 2020b; Nair & Finn, 2020; Wu et al., 2020; Caley et al., 2016). For example, Bansal et al. (2019) apply a CNN to the RGB input to predict the next waypoint—the ground truth of which is collected using trajectory optimization—then apply model-based planning. Active Neural SLAM (ANS) (Chaplot et al., 2020b) plans a path to the point goal (or a predicted long-term exploration goal) using a partial map of the environment, generating each subgoal to be within 0.25 m of the agent using an analytic shortest path planner. We stress that for navigation ANS does no global policy prediction; the PointGoal coordinates are simply fed in as the global goal.

The modular nature of these methods resonates with the proposed model. However, there are several important differences. First, we tackle AudioGoal, not PointGoal, which means our top-level module is not given the goal location and must instead learn how to direct the agent based on the audio inputs. Second, we introduce audio-visual subgoals; whereas visual subgoals focus on visible obstacle avoidance, audio-visual waypoints benefit from the wide reach of audio. For example, a visual subgoal may consider either of two exit doors as equally good, whereas an audio-visual subgoal prefers the one from which greater sound appears to be emerging. Third, a key novel element of our approach is to learn to generate navigation subgoals in an end-to-end fashion. In contrast, prior work relies on heuristics like selecting frontiers (Caley et al., 2016; Stein et al., 2018) or points along the shortest collision-free path (Bansal et al., 2019; Chaplot et al., 2020b) to define subgoals. Rather than use heuristics, our waypoints are directly predicted by the policy. This is a novel technical contribution to visual navigation independent of the audio-visual setting, as it frees the agent to dynamically identify subgoals driven by the ultimate navigation goal. Some recent work in hierarchical reinforcement learning (HRL) (Nachum et al., 2018; Li et al., 2019; Levy et al., 2019) explores ways to predict subgoals with a high-level policy end-to-end, but they train and test policies in the same environments with artificial low-dimensional state inputs, whereas we train our agent to generalize to unseen realistic 3D environments with visual and auditory sensory inputs.

**Visual semantic memory and mapping**  Learning-based visual mapping algorithms (Henriques & Vedaldi, 2018; Savinov et al., 2018; Gupta et al., 2017b;a) show exciting promise to overcome the limits of purely geometric maps. A learned map can encode semantic information beyond 3D points while being trained with the agent's ultimate task (like navigation). Recent work explores memories that spatially index learned RGB-D features (Tung et al., 2019; Henriques & Vedaldi, 2018; Gupta et al., 2017a), build a topological memory with visually distinct nodes (Savinov et al., 2018; Nagarajan et al., 2020; Chaplot et al., 2020a), or use attention models over stored visual embeddings (Fang et al., 2019). Expanding this line of work, we introduce the first multi-modal spatial memory. It encodes both visual and acoustic observations registered with the agent's movement along the ground plane. We show that the multi-modal memory is essential for the agent to produce good action sequences.

**Sound localization**  Robotics systems localize sound sources with microphone arrays (Nakadai & Nakamura, 1999; Rascon & Meza, 2017), and active control can improve localization (Nakadai et al., 2000; Wang et al., 2014). The geometry of a room can be in part sensed by audio, as explored with ideas for echolocation (Dokmanic et al., 2013; Christensen et al., 2020; Gao et al., 2020). In 2D video frames, methods learn to localize sounds based on their consistent audio-visual association (Hershey & Movellan, 2000; Tian et al., 2018; Senocak et al., 2018; Arandjelovic & Zisserman, 2018). Unlike

any of the above, we investigate the audio-visual navigation problem, where an agent learns to move efficiently towards a sound source in a 3D environment based on both audio and visual cues.

## 3 APPROACH

We consider the task of AudioGoal navigation (Chen et al., 2020; Gan et al., 2020). In this task the agent moves within a 3D environment and receives a sensor observation $O_t$ at each time step $t$ from its camera (depth) and binaural microphones. The environment is unmapped at the beginning of the navigation episode; the agent has to accumulate observations to understand the scene geometry while navigating. Unlike the common PointGoal task, for AudioGoal the agent does not know the location of the goal (*i.e.*, no GPS signal or displacement vector pointing to the goal is available). The agent must use the sound emitted by the audio source to locate and navigate successfully to the goal.

We introduce a novel navigation approach that predicts intermediate waypoints to reach the goal efficiently. Our approach is composed of three main modules (Fig. 2). Given visual and audio inputs, our model 1) encodes these cues using a perception and mapping module, then 2) predicts a waypoint, and finally 3) plans and executes a sequence of actions that bring the agent to the predicted waypoint. The agent repeats this process until it predicts the goal has been reached and executes the *Stop* action.

### 3.1 3D ENVIRONMENTS AND AUDIO-VISUAL SIMULATOR

We use the AI-Habitat simulator (Savva et al., 2019) with the publicly available Replica (Straub et al., 2019) and Matterport3D (Chang et al., 2017) environments together with the public SoundSpaces audio simulations (Chen et al., 2020). The 18 Replica environments are meshes constructed from real-world scans of apartments, offices, hotels, and rooms. The 85 Matterport3D environments are real-world homes and other indoor environments with 3D meshes and image scans.[1] The agent can travel through the spaces while receiving real-time egocentric visual and audio observations. Using SoundSpaces's room impulse responses (RIR), we can place an audio source in the 3D environment, then simulate realistic sounds at each location in the scene at a spatial resolution of 0.5m for Replica and 1m for Matterport3D. These state-of-the-art renderings capture how sound propagates and interacts with the surrounding geometry and surface materials, modeling all of the major features of the RIR: direct sound, early specular/diffuse reflections, reverberation, binaural spatialization, and frequency dependent effects from materials and air absorption (see Chen et al. (2020) for details). We experiment with 102 everyday sounds (details in Supp).

The simulator maintains a navigability graph of the environment (unknown to the agent). The agent can only move from one node to another if there is an edge connecting them and the agent is facing that direction. The action space $\mathcal{A}$ has four actions: *MoveForward*, *TurnLeft*, *TurnRight* and *Stop*.

We use these real-world image scans and highly realistic audio simulations to test our ideas in a reproducible evaluation setting. Please see the Supp video to gauge their realism. Our experiments further push the realism by considering distractor sounds and noisy sensors (cf. Sec. 4). We leave as future work to translate policies to a real-world robot, for which we are encouraged by recent sim2real attempts (Gupta et al., 2017a; Müller et al., 2018; Chaplot et al., 2020b; Stein et al., 2018).

### 3.2 PERCEPTION AND MAPPING

**Visual perception** At each time step $t$, we extract visual cues from the agent's first-person depth view, which is more effective for map construction than RGB (Chaplot et al., 2020b; Chen et al., 2019). First, we backproject the depth image into the world coordinates using the camera's intrinsic parameters to compute the local scene's 3D point cloud. Then, we project these points to a 2D top-down egocentric local occupancy map $L_t$ of size $3 \times 3$ meters in front of the agent, corresponding to the typical distance at which the real-world sensor is reliable. The map has two channels, one for the occupied/free space and one for explored/unexplored areas. A map cell is deemed occupied if it has a 3D point that is higher than 0.2m and lower than 1.5m, and it is deemed explored if any 3D point is projected into that cell (results are tolerant to noisy depth; see Supp). We update an allocentric geometric map $G_t$ by transforming $L_t$ with respect to the agent's last pose change and

---

[1]The AI2-THOR audio simulation from (Gan et al., 2020) is not yet available, and it contains synthetic computer graphics imagery (vs. SoundSpaces's use of real-world scans in Replica and Matterport).

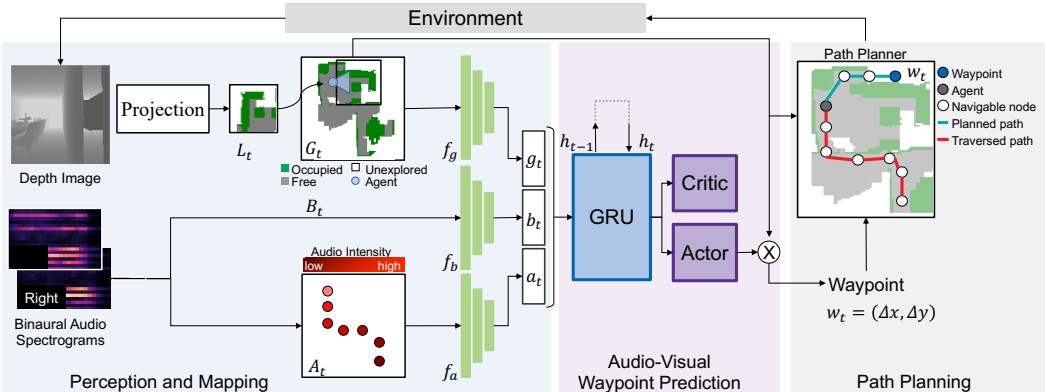

Figure 2: Model architecture. Our audio-visual navigation model uses the egocentric stream of depth images and binaural audio ($B_t$) to learn geometric ($G_t$) and acoustic ($A_t$) maps for the 3D environment. The multi-modal cues and partial maps (left) inform the RL policy's prediction of intermediate waypoints (center). For each waypoint, the agent plans the shortest navigable path (right). From this sequence of waypoints, the agent reaches the final AudioGoal efficiently.

then averaging it with the corresponding values of $G_{t-1}$. Cells with a value above $0.5$ are considered occupied or explored. See top branch in Figure 2.

**Acoustic perception** At each time step the agent receives binaural sound $B_t$ represented by spectrograms for the right and left ear, a matrix representation of frequencies of audio signals as a function of time (second branch in Figure 2; see Supp for spectrogram details). Beyond encoding the current sounds, we also introduce an *acoustic memory*. The acoustic memory is a map $A_t$ indexed on the ground plane like $G_t$ that aggregates the audio intensity over time in a structured manner. It records a moving average of direct sound intensity solely at positions visited by the agent. See the third branch in Figure 2. Note that a map of audio intensities reveals both distance and directional information about the sound source, since the gradient in audio intensity helps indicate the goal direction. The acoustic map and $B_t$ provide spatially grounded information about both the environment and the goal: the walls and other major surfaces influence the sound received by the agent at any given location, while the sound source at the goal gives a coarse sense of direction when the agent is far away. This directional cue gets increasingly precise as the agent approaches the goal.

### 3.3 AUDIO-VISUAL WAYPOINT PREDICTOR

Both the audio and visual inputs carry complementary information to set good waypoints en route to the audio goal. While the audio signals $B_t$ (binaural inputs) and $A_t$ (acoustic memory) inform the agent of the general direction of the goal and hint at the room geometry, the visual signal in the form of the occupancy map $G_t$ allows spatial localization of the waypoint and helps to avoid obstacles. Recall Figure 1, where the agent in the bedroom needs to reach a phone ringing in another room.

We learn three encoders to represent the inputs: $g_t = f_g(G_t)$, $b_t = f_b(B_t)$ and $a_t = f_a(A_t)$. Functions $f_g$ and $f_a$ first transform the geometric and acoustic maps ($G_t$ and $A_t$) such that the agent is located at the center of the map facing upwards and then crop them to size $s_g \times s_g$ and $s_a \times s_a$, respectively. Each function has a convolutional neural network (CNN) in the end to extract features (details in Supp). We concatenate the three vectors $g_t$, $b_t$ and $a_t$ to obtain the full audio-visual feature, and pass it into a gated recurrent neural network (GRU) (Chung et al., 2015). See Figure 2.

Our reinforcement learning waypoint predictor has an actor-critic architecture. It takes the hidden state $h_t$ of the GRU and predicts a probability distribution $\pi(W_t|h_t)$ over possible waypoints. $W_t$ is the action map of size $s_w \times s_w$ and represents the candidate waypoints in the area centered around the agent.[2] We mask the output of the policy with the local occupancy map to ensure that the model selects waypoints that are in free spaces. We sample a waypoint $w_t = (\Delta x, \Delta y)$ from $W_t$ according

---

[2]The environment graphs have nodes only where the SoundSpaces audio RIRs are available, and hence both actions and candidate waypoints are discrete sets. Note: this disallows testing noisy actuation for any method because audio observations are not available at positions off the RIR grid.

to the policy's predicted probability distribution. The waypoint is relative to the agent's current position and is passed to the planner (see Sec. 3.4).

This waypoint policy is an important element in our method design. It allows the agent to dynamically adjust its intermediate goals according to what it currently sees and hears. Unlike existing AV navigation methods, our waypoints guide the agent at a variable granularity, as opposed to fixing its actions to myopic next steps (Chen et al., 2020) or a final goal prediction (Gan et al., 2020). Unlike existing visual subgoal approaches, which rely on frontier-based heuristics or points along the shortest path (Chaplot et al., 2020b; Stein et al., 2018; Bansal et al., 2019; Caley et al., 2016), our waypoints are inferred in tight integration with the navigation task. Our results demonstrate the advantages.

## 3.4 PATH PLANNER

Given the generated waypoint $w_t$, a shortest-path planner tries to generate a sequence of low-level actuation commands chosen from $\mathcal{A}$ to move the agent to that waypoint. The planner maintains a graph of the scene based on the geometric map $G_t$ and estimates a path from the agent's current location to $w_t$ using Dijkstra's algorithm. Unexplored areas in the map are considered free space during planning (Chaplot et al., 2020b). Based on the shortest path, a low-level actuation command is analytically computed. The agent executes the action, gets a new observation $O_t$, updates both $G_t$ and $A_t$, and repeats the above procedure until it exits the planning loop.

The planning loop breaks under three conditions: 1) the agent reaches the waypoint, 2) the planner could not find a path to the waypoint (in this case the agent executes a random action before breaking the loop), or 3) the agent reaches a planning step limit. The planning step limit is set to mitigate bad waypoint prediction (due to noisy occupancy estimates) or hard-to-reach waypoints (like behind the wall of another room) from derailing the agent from the goal. If the model selects $w_t = (0, 0)$ (*i.e.*, the agent's current location), this means that the agent believes it has reached the final goal; the *Stop* action is then executed and the episode terminates.

## 3.5 REWARD AND TRAINING

Following typical navigation rewards (Savva et al., 2019; Chen et al., 2020), we reward the agent with $+10$ if it succeeds in reaching the goal and executing the *Stop* action there, plus an additional reward of $+0.25$ for reducing the geodesic distance to the goal and an equivalent penalty for increasing it. Finally, we issue a time penalty of $-0.01$ per executed action to encourage efficiency. For each waypoint prediction step, the agent is rewarded with the cumulative reward value collected during the last round of planner execution. Altogether, the reward encourages the model to select waypoints that are reachable, far from the current agent position, and on the route to the goal—or to choose the goal itself if it is within reach.

All learnable modules are jointly trained and updated every 150 waypoint prediction steps with Proximal Policy Optimization (PPO) (Schulman et al., 2017). The PPO loss consists of a value network loss, policy network loss, and an entropy loss to encourage exploration. Please see Supp for all implementation details.

## 4 EXPERIMENTS

**Environments** We test with SoundSpaces for Replica and Matterport environments in the Habitat simulator (Sec. 3.1). We follow the protocol of the SoundSpaces AudioGoal benchmark (Chen et al., 2020), with train/val/test splits of 9/4/5 scenes on Replica and 73/11/18 scenes on Matterport3D. We stress that the test and train/val environments are disjoint, requiring the agent to learn generalizable behaviors. Furthermore, for the same scene splits, we experiment with training and testing on disjoint sounds, requiring the agent to generalize to unheard sounds. For heard-sound experiments, the telephone ringing is the sound source; for unheard, we draw from 102 unique sounds (see Supp).

**Metrics** We evaluate the following navigation metrics: 1) success rate (SR), the fraction of successful episodes, i.e., episodes in which the agent stops exactly at the audio goal location on the grid; 2) success weighted by path length (SPL), the standard metric (Anderson et al., 2018) that weighs successes by their adherence to the shortest path; 3) success weighted by number of actions (SNA), which penalizes rotation in place actions, which do not lead to path changes. Please see Supp for more details on these comprehensive metrics.

Table 1: AudioGoal navigation results. Our audio-visual waypoints navigation model (AV-WaN) reaches the goal faster (higher SPL) and it is more efficient (higher SNA) compared to the state-of-the-art. SPL, SR, SNA are shown as percentages. For all metrics, higher is better. (H) denotes a hierarchical model.

| | Replica | | | | | | Matterport3D | | | | | |
| | Heard | | | Unheard | | | Heard | | | Unheard | | |
| Model | SPL | SR | SNA | SPL | SR | SNA | SPL | SR | SNA | SPL | SR | SNA |
|---|---|---|---|---|---|---|---|---|---|---|---|---|
| Random Agent | 4.9 | 18.5 | 1.8 | 4.9 | 18.5 | 1.8 | 2.1 | 9.1 | 0.8 | 2.1 | 9.1 | 0.8 |
| Direction Follower (H) | 54.7 | 72.0 | 41.1 | 11.1 | 17.2 | 8.4 | 32.3 | 41.2 | 23.8 | 13.9 | 18.0 | 10.7 |
| Frontier Waypoints (H) | 44.0 | 63.9 | 35.2 | 6.5 | 14.8 | 5.1 | 30.6 | 42.8 | 22.2 | 10.9 | 16.4 | 8.1 |
| Supervised Waypoints (H) | 59.1 | 88.1 | 48.5 | 14.1 | 43.1 | 10.1 | 21.0 | 36.2 | 16.2 | 4.1 | 8.8 | 2.9 |
| Gan et al. | 57.6 | 83.1 | 47.9 | 7.5 | 15.7 | 5.7 | 22.8 | 37.9 | 17.1 | 5.0 | 10.2 | 3.6 |
| Chen et al. | 78.2 | 94.5 | 52.7 | 34.7 | 50.9 | 16.7 | 55.1 | 71.3 | 32.6 | 25.9 | 40.1 | 12.8 |
| AV-WaN (**Ours**) (H) | **86.6** | **98.7** | **70.7** | **34.7** | **52.8** | **27.1** | **72.3** | **93.6** | **54.8** | **40.9** | **56.7** | **30.6** |

**Existing methods and baselines**  We compare the following methods (detailed in Supp; see Tab. 3):

– **Random**: an agent that randomly selects each action and signals *Stop* when it reaches the goal.

– **Direction Follower**: a hierarchical model that sets intermediate goals $K$ meters away in the audio's predicted direction of arrival (DoA), and repeats. $K$ is estimated through a hyperparameter search on the validation split, which yields $K = 2$ in Replica and $K = 4$ in Matterport. We train a separate classifier based on audio input to predict when this agent should stop.

– **Frontier Waypoints**: a hierarchical model that intersects the predicted DoA with the frontiers of the explored area and selects that point as the next waypoint. Frontier waypoints are commonly used in the visual navigation literature, e.g., (Caley et al., 2016; Stein et al., 2018; Chaplot et al., 2020b), making this a broadly representative baseline for standard practice.

– **Supervised Waypoints**: a hierarchical model that uses the RGB frame and audio spectrogram to predict waypoints in its field of view (FoV) with supervised (non-end-to-end) learning. This model is inspired by Bansal et al. (2019), which learns to predict waypoints in a supervised fashion.

– **Chen et al. (2020)**: a state-of-the-art end-to-end AudioGoal RL agent that selects actions using audio-visual observations. It lacks any geometric or acoustic maps. We run the authors' code.

– **Gan et al. (2020)**: a state-of-the-art AudioGoal agent that predicts the audio goal location from binaural spectrograms alone and then navigates with an analytical path planner on an occupancy map it progressively builds by projecting depth images. It uses a separate audio classifier to stop. We adapt the model to improve its performance on Replica and Matterport, since the authors originally tested on a game engine simulator (see Supp).

**Navigation results**  We consider two settings: 1) *heard sound*—train and test on the telephone sound, following (Chen et al., 2020; Gan et al., 2020), and 2) *unheard sounds*—train and test with disjoint sounds, following (Chen et al., 2020). In both cases, the test environment is always unseen, hence both settings require generalization.

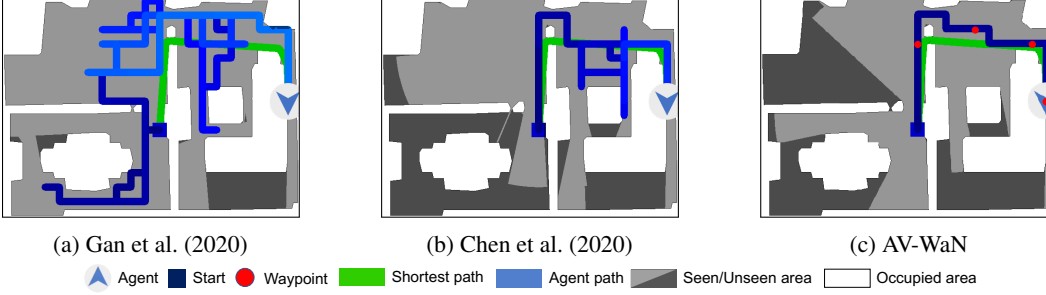

|  (a) Gan et al. (2020) |  (b) Chen et al. (2020) |  (c) AV-WaN |
|---|---|---|

▲ Agent  ■ Start  ● Waypoint  ▬ Shortest path  ▬ Agent path  ▬ Seen/Unseen area  ☐ Occupied area

Figure 3: Navigation trajectories on top-down maps vs. all existing AudioGoal methods. Agent path fades from dark blue to light blue as time goes by. Green is the shortest geodesic path in continuous space. All agents have reached the goal. Our waypoint model navigates to the goal more efficiently. The agent's inputs are egocentric views (Fig. 1); figures show the top-down view for ease of viewing the full trajectories.

Table 2: Ablation study for AV-WaN. Results are averaged over 5 test runs; all standard deviations are $\leq 0.5$.

| | Replica | | | | | | Matterport3D | | | | | |
| | Heard | | | Unheard | | | Heard | | | Unheard | | |
| Model | SPL | SR | SNA | SPL | SR | SNA | SPL | SR | SNA | SPL | SR | SNA |
|---|---|---|---|---|---|---|---|---|---|---|---|---|
| AV-WaN w/o $A_t$ and $G_t$ | 84.3 | 97.8 | 69.1 | 34.0 | 48.6 | 25.4 | 68.8 | 92.1 | 52.1 | 20.5 | 30.4 | 15.5 |
| AV-WaN w/o $G_t$ | 85.1 | 97.5 | 69.0 | 27.0 | 45.6 | 20.3 | 70.2 | 94.0 | 52.4 | 25.4 | 45.0 | 19.2 |
| AV-WaN w/o $A_t$ | 85.7 | 98.7 | 70.2 | 34.5 | **63.3** | 24.8 | 70.2 | 93.6 | 53.2 | 36.7 | 53.8 | 28.6 |
| AV-WaN w/o waypoints | 79.8 | 95.5 | 48.4 | 25.5 | 38.2 | 10.6 | 44.3 | 63.2 | 20.3 | 25.5 | 40.0 | 11.0 |
| AV-WaN | **86.6** | **98.7** | **70.7** | **34.7** | 52.8 | **27.1** | **72.3** | **93.6** | **54.8** | **40.9** | **56.7** | **30.6** |

Table 1 shows the results. We refer to our model as AV-WaN (**A**udio-**V**isual **Wa**ypoint **N**avigation). Random does poorly due to the challenging nature of the AudioGoal task and the complex 3D environments. For the heard sound, AV-WaN strongly outperforms all the other methods—with 8.4% and 29% SPL gains on Replica compared to Chen et al. and Gan et al., and 17.2% and 49.5% gains on Matterport. This result shows the advantage of our dynamic audio-visual waypoints and structured acoustic map, compared to the myopic action selection in Chen et al. and the final-goal prediction in Gan et al. We find that the RL model of Chen et al. fails when it oscillates around an obstacle. Meanwhile, predicting the final audio goal location, as done by Gan et al., is prone to errors and leads the agent to backtrack or change course often to redirect itself towards the goal. This result emphasizes the difficulty of the audio-visual navigation task itself; simply reducing the task to PointGoal after predicting the goal location from audio (as done in Gan et al.) is much less effective than the proposed model. See Figure 3.

Our method also surpasses all three other hierarchical models. This highlights our advantage of directly *learning* to set waypoints, versus the heuristics used in current hierarchical visual navigation models. Even the Supervised Waypoints model does not generalize as well to unseen environments as AV-WaN. We expect this is due to the narrow definition of the optimal waypoint posed by supervision compared to our model, which learns from its own experience what is the best waypoint for the given navigation task in an end-to-end fashion.

In the unheard sounds setting covering 102 sounds (Table 1, right), our method again strongly outperforms all existing methods on both datasets and in almost every metric. The only exception is our 2.8% lower SPL vs. Chen et al. on Replica, though our model still surpasses Chen et al. in terms of SNA on that dataset, meaning we have better accuracy when normalizing for total action count. Absolute performance declines for all methods, though, due to the unfamiliar audio spectrogram patterns. The acoustic memory is critical for this important setting; it successfully abstracts away the specific content of the training sounds to better generalize.

**Ablations** Table 2 shows ablations of the input modalities and the audio-visual waypoint component of our model.[3] Removing both the geometric and acoustic maps causes a reduction in performance. This is expected since without $A_t$ and $G_t$, the model has only the current audio observation $B_t$ to predict the next waypoint. Notably, even this heavily ablated version of our model outperforms the best existing model (Chen et al., 2020) (see Table 1). This shows that our waypoint-based navigation framework itself is more effective than the simpler RL model (Chen et al., 2020), as well as the existing subgoal approaches. Removing just $A_t$ also leads to a drop in performance, which demonstrates the importance of the proposed structured acoustic memory for efficient navigation. Both $A_t$ and $G_t$ are complementary and critical for our model to reach its best performance. Finally, we evaluate the impact of our idea of audio-visual waypoint prediction. We replace the actor network in our model (see Fig. 2 middle) with a linear layer that outputs the action distribution over the four primitive actions in $\mathcal{A}$. An action sampler directly samples an action from this distribution and executes it in the environment. In this case, there is no need for a planner. Our gains over that ablation confirm the value of the waypoints to our model, even when all other components are fixed.

**Failure cases** We next analyze the unsuccessful episodes for our model (see Supp video for examples). We identify two repeating types of failures among these episodes. The first is where the audio goal is cornered among obstacles or lies right next to a wall. In this case, while AV-WaN reaches the goal quickly, it keeps oscillating around it and fails to pinpoint the location of the goal due to strong audio

---

[3]When $G_t$ is removed, we remove the masking operation to ensure no geometric information is used as input, but we keep the geometric map for the planner.

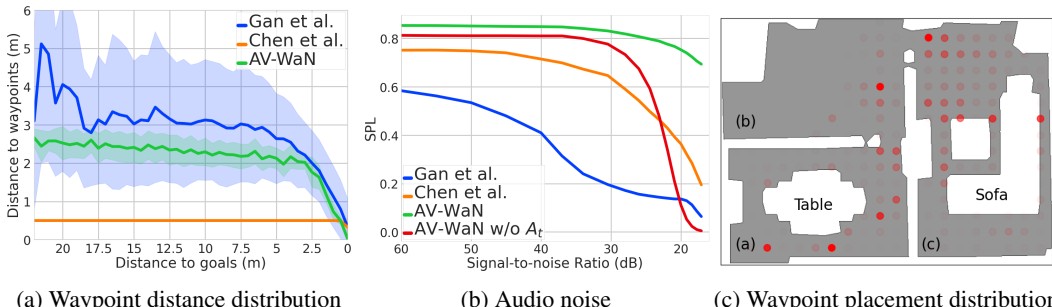

(a) Waypoint distance distribution      (b) Audio noise      (c) Waypoint placement distribution

Figure 4: Analysis of selected waypoints (a,c) and accuracy vs. microphone noise (b). See text.

reflections from the obstacles around the goal or due to mapping errors. In the second case, we notice that sometimes the agent prematurely executes a stop action next to the audio goal. We expect that the differences in the audio intensity in the immediate neighborhood of the goal where the sound is the loudest are harder to detect, which may lead to this behavior.

**Noisy audio and distractor sounds** To understand the robustness of our model under noisy audio perception, we consider two sources of audio noise: environment noise and microphone noise. For environment noise, we add distractor sounds (e.g. human speaking, fan spinning) to interfere with the agent's audio perception. The agent is always tasked to find the telephone, and the distractor is an unheard sound placed at a random location. At each time step, the agent receives the combined waveforms of two sounds and needs to pick up on the telephone signal and find its source location. We use the same episodes from the Heard experiment (Table 1) to train and evaluate the agent. With distractors, the best performing baseline, Chen et al., obtains $71.7\%$ and $53.3\%$ test SPL on Replica and Matterport respectively, while our model achieves $83.1\%$ and $70.9\%$. For microphone noise, we add increasing Gaussian noise to the received audio waveforms. Fig. 4b shows the results. AV-WaN is quite robust to audio noise, especially with $A_t$, while the existing AudioGoal methods suffer significantly. Hence our model's advantages persist in noisier settings common in the real world, and the acoustic memory is essential in this noisy setting.

**Dynamic waypoint selection** Fig. 4a plots the distribution of euclidean distances to waypoints as a function of the agent's geodesic distance to the goal. We see that our agent selects waypoints that are further away when it is far from the goal, then predicts closer ones when converging on the goal. Please see Supp for details and analysis.

**Placement of waypoints** To examine how waypoints are selected based on surrounding geometry, Fig. 4c plots the distribution of waypoints on a top-down map for a test Replica environment. The waypoints are accumulated over trajectories with start or end points in room $a$ or room $c$, and goal locations are excluded. We see waypoints are mostly selected around obstacles and doors, which are the decision states that lie at critical junctions in the state spaces from which the agent can gather the most new information and transition to new, potentially unexplored regions (Goyal et al., 2019). The most frequent waypoints are usually 2-3m apart, close to the maximum distance the agent can choose.

## 5    CONCLUSION

We introduced a reinforcement learning framework that learns to set waypoints for audio-visual navigation with an acoustic memory. Our method improves the state of the art on the challenging AudioGoal problem, and our analysis shows the direct impact of the new technical contributions. In future work we plan to consider AV-navigation tasks of increasing complexity, such as semantic sounds, moving sound sources, and real-world transfer.

## ACKNOWLEDGEMENTS

UT Austin is supported in part by DARPA Lifelong Learning Machines and ONR PECASE N00014-15-1-2291.

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

# 6 SUPPLEMENTARY MATERIAL

In this supplementary material we provide additional details about:

- Video (with audio) for qualitative assessment of our agent's performance.
- Implementation details (Sec. 6.2)
- Noisy depth experiment (Sec. 6.3)
- Binaural spectrogram calculation details (Sec. 6.4).
- Details of the CNN encoders from the perception and mapping component of our AV-WaN (Sec. 6.5).
- Details on the navigation metric definitions (Sec. 6.6).
- Baseline implementation details (Sec. 6.7).
- Details on the dynamic waypoint selection analysis (Sec. 6.8).
- Details on the unheard sounds data splits (Sec. 6.9).

## 6.1 QUALITATIVE VIDEO

The supplementary video demonstrates the audio simulation platform that we use and shows the comparison between our proposed model and the baselines as well as qualitative results from the unheard and time-varying sound experiments. Please listen with headphones to hear the binaural audio correctly.

## 6.2 IMPLEMENTATION DETAILS

We train our model with Adam (Kingma & Ba, 2014) with a learning rate of $2.5 \times 10^{-4}$. The output of the three encoders $g_t$, $b_t$ and $a_t$ are all of dimension 512. We use a one-layer bidirectional GRU (Chung et al., 2015) with 512 hidden units that takes $[g_t, b_t, a_t]$ as input. The geometric map size $s_g$ is 200 at a resolution of 0.1m. The acoustic map size $s_a$ and the action map size $s_w$ are 20 and 9 respectively, at the same resolution as the environment. We use an entropy loss on the policy distribution with coefficient 0.02. We train for 7.5 million policy prediction steps, and we set the upper limit of planning steps to 10.

## 6.3 NOISY DEPTH PERCEPTION

Similar to our experiments on the model robustness against noisy audio perception, we test here the impact of noise on depth, the other input modality used by the agents. We use the standard Redwood depth noise model from Choi et al. (2015). To imitate a Sim2Real scenario, we do not retrain the models with the noisy depth sensor; we use noisy depth at inference time only. We find that all models are robust to this type of noise, with SPL performance varying by less than 1% and SR by less than 0.5%. These mild reductions for all methods are smaller than the margins separating the different baselines, meaning the conclusions from the main results hold with noisy depth at test time.

## 6.4 SPECTROGRAM DETAILS

Following Chen et al. (2020), we first compute the Short-Time Fourier Transform (STFT) with a hop length of 160 samples and a windowed signal length of 512 samples, which corresponds to a physical duration of 12 and 32 milliseconds at a sample rate of 44100Hz (Replica) and 16000Hz (Matterport). STFT gives a $257 \times 257$ and a $257 \times 101$ complex-valued matrix respectively for one second audio clip; we take its magnitude, downsample both axes by a factor of 4 and take the logarithm. Finally, we stack the left and right audio channel matrices to obtain a $65 \times 65 \times 2$ and a $65 \times 26 \times 2$ tensor.

A room impulse response (RIR) is characterized by three stages: direct sound, early reflections, and reverberation. Direct sound is a strong signal of agent's distance to goal. We compute the intensity of the direct sound part of the audio signal by taking the root-mean-square (RMS) value of the first 3ms non-zero audio waveform averaged between the left and right channels.

Table 3: Summary of methods. I(A, B) denotes the intersection of A and B. The DoA predictor and the stopping function in the Direction Follower and Frontier waypoints are trained but their waypoints are computed analytically. In constrast, Supervised Waypoints trains its waypoint predictor with supervised learning (SL).

| | | Model | | | Waypoints | | |
|---|---|---|---|---|---|---|---|
| Model | Hierarchical | Granularity | Learned | Definition | Learning | Dynamic |
| Random | ✗ | Primitive action | ✗ | - | - | - |
| Chen et al. | ✗ | Primitive action | ✓ | - | - | - |
| Gan et al. | ✗ | Final goal | ✓ | - | - | - |
| Direction Follower | ✓ | Waypoint | ✓ | $K$ steps in DoA | - | ✗ |
| Frontier Waypoints | ✓ | Waypoint | ✓ | I(frontier, DoA) | - | ✗ |
| Supervised Waypoints | ✓ | Waypoint | ✓ | I(shortest path, FoV) | SL | ✗ |
| AV-WaN (**Ours**) | ✓ | Waypoint | ✓ | Learned end-to-end | RL | ✓ |

## 6.5 CNN ARCHITECTURE DETAILS

The CNN component of the $f_g$ and $f_b$ encoders has three convolution layers each, with kernel sizes of [8, 4, 3] and strides of [4, 2, 1] respectively. Similarly, the CNN component of $f_a$ has three convolution layers with kernel sizes of [5, 3, 3] and strides of [2, 1, 1]. For all CNNs, the channel size doubles after each convolution layer starting from 32 and each convolution layer is followed by a ReLU activation function. We use a fully connected layer at the end of each CNN to transform the CNN features into an embedding of size 512.

## 6.6 METRIC DEFINITIONS

Next we elaborate on the navigation metrics defined in Sec. 4 of the main paper.

1. Success Rate (SR): the fraction of successfully completed episodes, *i.e.*, the agent reaches the goal within the time limit of 500 steps and selects the stop action exactly at the goal location.

2. Success weighted by path length (SPL) (Anderson et al., 2018): weighs the successful episodes with the ratio of the shortest path $l_i$ to the executed path $p_i$, SPL $= \frac{1}{N} \sum_{i=1}^{N} S_i \frac{l_i}{\max(p_i, l_i)}$.

3. Success weighted by number of actions (SNA): weighs the successful episodes by the ratio of the number of actions taken for the shortest path $l_i^a$ to the number of executed actions by the agent's $p_i^a$, SNA $= \frac{1}{N} \sum_{i=1}^{N} S_i \frac{l_i^a}{\max(p_i^a, l_i^a)}$. This metric captures the agent's efficiency in reaching the goal. Two agents may have different number of actions but the same SPL score for an episode since the number of actions also accounts for actions that do not lead to path changes, like rotation in place.

## 6.7 BASELINE IMPLEMENTATION DETAILS

Table 3 summarizes the properties of the methods we compare to our AV-WaN model. Note that three of the compared methods are also hierarchical navigation models that modularly combine subgoal setting with low-level planning or navigation (left). As noted in the text, the key distinction is in how the waypoints are set (right). In particular, our model is the only hierarchical navigation model that learns a policy to set the waypoints with reinforcement learning, end-to-end with the navigation task.

Next we provide further implementation details about the baselines and existing methods. Note that all hierarchical models and Gan et al. (2020) share the same mapping and planning modules.

### 6.7.1 GAN ET AL. (2020)

Since code for (Gan et al., 2020) was not available at the time of our submission, we implemented this method ourselves. We followed instructions given by the authors, and also implemented our own enhancements to improve its performance on the the Replica and Matterport (MP3D) datasets.

We use a VGG-like CNN to predict the relative location $(\Delta x, \Delta y)$ of the audio goal given the binaural audio spectrograms as input. The CNN has 5 convolutional (conv.) layers interleaved with 5 max pooling layers and followed by 3 fully-connected (FC) layers. Each of the conv. and FC layers has a batch-normalization (Ioffe & Szegedy, 2015) of $10^{-5}$ and a ReLU activation function except for

the last FC layer which outputs the prediction. The conv. layers have the following configuration - a square kernel of size 3, a stride of 1 in both directions, and a symmetric zero-padding of 1. The number of output channels of the 5 conv. layers are {64, 128, 256, 512, 512} in order. The max pooling layers have a square kernel of size 2 with a stride of 2 in each direction except for a stride of 1 in the last max pooling layer for MP3D experiments. The FC layers have sizes {128, 128, 2} in order. We train the network until convergence to lower the minimum squared error (MSE) loss using Adam (Kingma & Ba, 2014) with an initial learning rate of $3 \times 10^{-3}$ and a batch size of 128 and 1024 for Replica and MP3D respectively. The model from (Gan et al., 2020) has a separate audio classifier for stopping. This classifier has the same architecture as the goal prediction model except for last FC layer which just has 1 output unit and a sigmoid activation. The stopping classifier is trained to minimize the binary cross entropy (BCE) loss with an initial learning rate of $3 \times 10^{-5}$.

During navigation, the agent predicts the AudioGoal location after every $N$ time steps if the predicted location is not reached before that. The agent stops if the episode times out after 500 time steps or the stopping classifier predicts the stop action. The original paper sets $N = 1$ but we found that 1-step predictions are very reactive in nature, *i.e.*, the agent keeps going back and forth or keeps turning while standing at the same location. This leads to a very low performance in the realistic Replica and MP3D test environments (Chen et al., 2020). We improve the prediction stability and the navigation performance by predicting after every $N$ steps where $N$ is chosen through a hyperparameter search on the validation split. For Replica, $N$ is set to 20 for both heard and unheard sounds, while for MP3D, $N$ is set to 50 and 60 for heard and unheard sounds respectively.

### 6.7.2  DIRECTION FOLLOWER

For this baseline, we train a CNN model to predict the direction of arrival (DoA) of the sound with the binaural spectrograms as input. We collect the ground truth DoAs by using the ambisonic room impulse responses (RIR) sampled at 44.1 kHz for Replica and 16.0 kHz for MP3D from (Chen et al., 2020). The first sound samples from the RIRs that correspond to the direct sound are used to build a circular intensity map around the agent at the height of the agent's ears. The circular map is discretized into 36 bins where each bin is equal to $360°/36 = 10°$. We select the bin with the maximum intensity in this map to approximate the DoA of the direct sound.

We use the exact same VGG-like architecture from the (Gan et al., 2020) re-implementation except for replacing the output layer with a single fully-connected layer with 36 output units for classifying a binaural spectogram pair into one of the 36 classes where each class corresponds to a $10°$ DoA bin. The network is trained until convergence to lower the negative log-likelihood (NLL) loss with Adam (Kingma & Ba, 2014), a batch size of 128 and for 1024 for Replica and MP3D respectively, and an initial learning rate of $3 \times 10^{-4}$ for both heard and unheard sounds for both the environments.

During navigation in both Replica and MP3D environments, the agent predicts the DoA using the previous model and moves to an intermediate goal that is 4 steps away (2 meters) in that direction. The value of 4 is chosen through a hyperparameter search using the validation split. The intermediate goal is recomputed after every 4 time steps as long as the agent has not reached the audio goal. If the predicted intermediate goal does not lie at a navigable location in the agent's geometric map ($G_t$), it executes a random action. The agent uses the same audio classifier as the Gan et al. (2020) baseline for stopping.

### 6.7.3  FRONTIER WAYPOINTS

Similar to the Direction Follower baseline, this agent also predicts the DoA of the direct sound but moves to the nearest frontier (Caley et al., 2016; Stein et al., 2018) in that direction instead of an intermediate goal that is four steps away. To improve this baseline, we enforce an additional constraint so that the frontier point is always at least 3 steps (1.5 meters) away to ensure that the agent does not make reactive predictions and keep going back and forth between the same two points in the environment. If there is no frontier point along the predicted DoA (a common case when the agent starts off), then the agent simply moves 3 steps in that direction. If the agent finds the next frontier to be $N$ steps away, then the agent does not predict another frontier waypoint until it reaches the current one or $2N$ time steps have passed. For stopping, the agent uses the same stopping classifier as the previous baselines.

### 6.7.4 SUPERVISED WAYPOINTS

This baseline is a hierarchical model that predicts waypoints in its field of view (FOV) with supervised (non-end-to-end) learning. This model is inspired by Bansal et al. (2019) where they use an expert policy to generate ground truth waypoints and train a visual waypoint predictor based on RGB inputs and point goal information. Similarly, in this baseline we use RGB observations with the audio goal spectrograms to train a model to predict waypoints where the ground truth for these waypoints is generated using a shortest path planner.

Specifically, in this implementation, we use a similar audio encoder as in the other baselines. For encoding the RGB images, we use the exact same architecture from $f_g$ other than modifying the first convolution layer to accommodate for the three RGB channels. We do a mid-level fusion of the outputs of these two convolutional encoders and the fused output is fed to FC layers which have the same sizes as the FC layers in the (Gan et al., 2020) model. The ground truth target for the model is the relative location $(\Delta x, \Delta y)$ of the point of intersection of the shortest path to the audiogoal location and a local field of view (FoV) around the agent. For all our experiments, we choose a 8m $\times$ 8m square with the agent at the center as the FoV. We train this model using the same hyperparameters as the (Gan et al., 2020) baseline. For stable prediction and better navigation performance, the navigation agent repredicts after every 15 and 25 steps for heard and unheard sounds in Replica, and after every 30 steps for both sound types in MP3D. This hyperparameter is again estimated through a hyperparameter search on the validation split.

### 6.8 DYNAMIC WAYPOINT SELECTION DETAILS

To analyze the behavior of dynamic waypoint selection, Fig. 4a plots the distribution of euclidean distances to waypoints as a function of the agent's geodesic distance to the goal collected from all prediction steps across all episodes on Replica. The existing methods do not set waypoints. For Chen et al. (2020), we say the "waypoint" distance is 0m if the agent chooses to stop and 0.5m otherwise. For Gan et al. (2020), we say "waypoints" are the intersection of the predicted vector to the final goal and the explored area.

We see that our agent selects waypoints that are further away when it is far from the goal, then predicts closer ones when converging on the goal. In contrast, the step-by-step model (Chen et al., 2020) effectively has a fixed radius waypoint, while the final-goal model (Gan et al., 2020) has a high variance even when close to the goal, indicative of the redirection and backtracking behavior described above. The large waypoint distances for (Gan et al., 2020) are a symptom of its backtracking due to misprediction and lack of temporal modeling; since depth projections are limited to 3m, an ideal agent should always pick a waypoint up to 3m away and push forward to the goal.

### 6.9 UNHEARD SOUNDS DATA SPLITS

Following Chen et al. (2020), we utilize 102 copyright-free natural sounds across a wide variety of categories: air conditioner, bell, door opening, music, computer beeps, fan, people speaking, telephone, and etc. These 102 sounds are divided into non-overlapping 73/11/18 splits for train, validation and test.

In Table 1, for the Heard sound experiment, we use the sound source of 'telephone'. For the Unheard sound experiment, we use the 78 sounds for training scenes, and generalize to unseen scenes as well as unheard sounds. Particularly, we utilize the 11 sounds for validation scenes, and the remaining 18 sounds for test scenes.

