# OpenReview forum: "Learning to Set Waypoints for Audio-Visual Navigation"
_ICLR.cc/2021/Conference — ICLR 2021 Poster_

### Official Review · AnonReviewer2 · 2020-10-17
**Interesting submission on audio-visual navigation. The role of waypoints can be better justified.**

**Rating:** 6
**Confidence:** 4

**Review:**

This paper studies the problem of navigating to the sound source in a virtual environment such as Replica.  The main contribution is a new formulation that learns a policy on the next "waypoint" and uses the predicted waypoints as intermediate goals for path planning. The results are promising, much better than recently published baselines especially Gan et al and Chen et al.

The paper is complete, well-written, and the reference is thorough. The formulation is well-justified. The results look promising. The authors have also included some interesting analyses to better understand how the model works.

On the negative side, I'm still not fully convinced that this is a practically useful problem, or how challenging it could be if formulated in the right way: assuming a household robot has a room-centric representation once deployed, even simply walking through all rooms will let it quickly identify the audio source. In this case, it's unclear why we need such complicated policy learning algorithms. But I don't mean to reject this paper based on this philosophical argument.  The authors don't have to respond to this point.

What I do want to hear from the authors is why waypoints are useful. Though the authors have included some ablated models, there misses one baseline that employs exactly the current formulation but lets the policy learn to predict the next step (action) directly instead of to predict waypoints. Path planning is therefore no longer required. If the authors restrict the action space to be the space of rooms (i.e. actions are "go to the room on the right", instead of "moving right by 1 foot"), unless the agent believes it's already in the same room as the audio source, then such a policy learning method may work quite well, maybe even comparable with the waypoint-based method?

---

> ### Author Response · Authors · 2020-11-20
> **Response to reviewer 2**
>
> We appreciate your helpful feedback.
>
> 1: Missing one baseline that predicts next step directly instead of waypoint.
> As suggested by the reviewer, we have experimentally validated that the waypoints are better than next-step actions by training our model to predict low-level actions directly instead of waypoints, and updated the 'Ablations' paragraph in Section 4 of the main paper with the results. We note that our full AV-WaN model outperforms this ablation by a large margin. For example, on Matterport3D our model achieves 28\% and 15\% higher SPL compared to the ablated version without waypoints for the heard and unheard sound settings, respectively.
>
> 2: Search for audio goal using a room-centric representation.
> Recall that the AudioGoal task requires the agent to navigate in unmapped environments (e.g., rescue search) as is the case in other navigation tasks like PointGoal and ObjectGoal. Lacking a floor plan at the start of the episode makes it hard for the agent to identify the rooms' layout. Furthermore, our model is flexible in terms of where to set the next waypoint. If the model finds strong cues to enter a room based on its observations then it will set waypoints that will take it towards the door of the room and then inside (as the agent gradually builds the map of the environment).
>
> Finally, a brute-force room search could be applicable in one- or two-room apartments (although the agent still needs to find the exact location of the goal, see Fig. 3), but it is highly inefficient in large environments like Matterport3D (Chang et al., 3DV 2017) where one scene has more than 22 rooms on average.

---

### Official Review · AnonReviewer3 · 2020-10-28
**Official Blind Review #3**

**Rating:** 7
**Confidence:** 4

**Review:**

Summary:

The authors address the audio-visual navigation problem, which aims to find a sound source in a 3D environment using both audio and visual information. The key innovation of the paper is to learn to set audio-visual waypoints, which decomposes a final goal to useful subgoals. Acoustic memory is introduced to strengthen the auditory perception. Experiments are performed on 3D environments of Replica and Matterport3D using SoundSpaces audio.

Pros:

(1)  A deep reinforcement learning approach for AudioGoal navigation with audio-visual waypoints is proposed. It learns to set useful subgoals and address the navigation in a hierarchical manner.

(2) The experiments are thorough and can well validate the effectiveness of the proposed audio-visual waypoint-based approach.

(3) The paper is easy to follow and the provided demo can nicely illustrate the problem and demonstrate the superiority of the proposed method.

Cons:

(1) Rather than only current audio, the authors propose to use the acoustic memory, which aggregates the audio intensity over time in a structured manner. Although the authors claim that acoustic memory can strengthen auditory perception, we only observe relatively small improvements (AV-WaN vs. AV-WaN w/o At) in Table 2.

(2) Any failure examples? Please provide some failure results of the proposed audio-visual waypoint-based method and give an analysis in the main paper. Failure cases can help us to understand the drawbacks of the current subgoal based model.

*** Post-Rebuttal ***

The authors addressed my concerns in the rebuttal. Overall, this is an interesting paper and extensive experiments are conducted. Thus, I would like to keep my positive rating.

---

> ### Author Response · Authors · 2020-11-20
> **Response to reviewer 3**
>
> We appreciate your positive feedback.
>
> 1: Acoustic memory gives relatively small improvements.
> The acoustic map gives relatively small improvements when used with clean audio. However, when used with noisy audio, our model with an acoustic map is much more robust than the other models, as can be seen in Fig. 4b. Specifically, when the noise level exceeds 30 dB, our model with the acoustic memory suffers a very minor decline in performance; however, without the acoustic memory we see the noise has a significant impact on the model.
>
> 2: Any failure examples?
> Yes, we provided some failure examples in the supplementary video (see starting from 5:25). Sometimes when the audio goal is just next to a wall or cornered between obstacles, the audio reflections could be strong, and the agent after reaching the goal quickly would oscillate around the goal trying to locate the exact location. We also saw some cases where the agent would issue a stop action prematurely just next to the goal. We expect the changes in audio intensity are less detectable in the immediate area around the goal where the audio is the loudest which may lead to this behavior. We have updated the paper to include an analysis of failure cases in `Navigation results' under Section 4.

---

### Official Review · AnonReviewer4 · 2020-10-28
**Recommendation to Accept**

**Rating:** 7
**Confidence:** 3

**Review:**

This paper tackles the AudioGoal task of navigating to a acoustic source in a 3D environment. It introduces the idea of an acoustic memory, which maps and aggregates acoustic intensity over time. An agent’s acoustic memory, in tandem with its egocentric depth view, is then used to select navigation waypoints in an end-to-end manner. Their method beats SoTA in AudioGoal for two environments: Replica and Matterport3D.

The paper presents a simple end-to-end solution to waypoint selection that sits on top of an environment’s low-level controls. There’s a very nice symmetry between the occupancy map (used for waypoint selection) and the acoustic memory map — backed, of course, by experimental results and convincing ablations.

Overall, the paper is extremely well written. Namely, in the exposition of the AudioGoal task. This is coming from someone who (works on embodied language and) is aware of, but not deeply familiar with, the tasks.

The paper provides a comprehensive set of experiments, baselines, and ablations. I particularly like Figure 4, which demonstrates the efficacy of acoustic memory in the presence of microphone noise.

Finally, most clarifying questions that I had are addressed in the Supplemental section — not distracting from the main points of the paper.

[Clarifying Questions]

In Section 3.5, the authors should define what a successful episode means in each respective environment (e.g., within 3 meters of the goal). This affects how SR and SPL are interpreted.

[Post Rebuttal]

Thank you to the authors for addressing my question. The paper presents a simple and elegant approach to the AudioGoal task, backed by extensive experiments and good writing. I'd like to maintain my positive rating.

---

> ### Author Response · Authors · 2020-11-20
> **Response to reviewer 4**
>
> We appreciate your positive feedback.
>
> Definition of successful episode.
> An episode is successful if and only if the agent stops at the exact audio goal location on the grid, as mentioned in the definition of Success Rate in Section 6.7 in the supplementary. We have also included the criterion for success in 'Metrics' in Section 4 of the revised main paper.

---

### Official Review · AnonReviewer1 · 2020-11-03
**Thorough and useful paper, with some clarifications and restated contributions.**

**Rating:** 7
**Confidence:** 4

**Review:**

This work presents an approach for audio-visual navigation, in which an agent receives both an RGBD observation of the world and an audio signal emitted from the goal. The proposed approach leverages a structured memory via an occupancy grid and an acoustic map. A learned hierarchical policy is used to set waypoints within the occupancy grid at a high level, with a low level search over the free occupancy grid. The approach is demonstrated over baselines to reach the goal at a high rate and to do so efficiently.

The paper is well written and clear. The figures and videos are useful. The baselines and results are thorough and show clear benefit of the method and design choices. I appreciate both the comparison to state of the art methods for audio-visual and the baseline comparisons. A few clarifications that should be made:
- The right side of Fig. 2 is slightly unclear due to the graph, which on a quick look brings notions of techniques like Savinov 2018. As the graph is just used by the simulator, I’m not sure it makes sense to visualize in this way.
- The figures alternate between showing the observation as RGB and as depth. My understanding from text is that this uses RGB-D, but from figures like Fig. 2 it is not clear where the RGB is used. For Fig. 1 the depth is not shown (though from reading, I understand it to be projected into the occupancy map).
- Is directionality from the audio signal used at all within the acoustic memory?
- What happens if the waypoint is not possible with the graph search?
- How are unexplored regions treated for the graph search?

The paper is somewhat limited by the impactfulness of the setting, audio-visual navigation. The authors make a clear case for uses of such a problem, but in general the setting appears somewhat manufactured. It boils down to a setting like point navigation but with a noisily observed goal with an uncertainty distribution based on audio. Another setting with this noisy goal is something like semantic or object navigation, e.g., https://arxiv.org/pdf/2007.14545.pdf, https://arxiv.org/pdf/2007.00643.pdf. Overall I believe approaches from this work may be applied in these settings and the paper could have significantly greater impact if these settings were considered. At a minimum, I believe the paper would benefit from a discussion of applications of ideas from this work beyond audio-visual navigation.

My other concern is that at times the paper is unclear or overstates contributions. Such as stating:
- “This is a novel technical contribution independent of the audio-visual setting, as it frees the agent to dynamically identify subgoals driven by the ultimate navigation goal.” and “This is a new idea for 3D navigation subgoals in general, not specific to audio-visual”. Many of the cited navigation papers use a hierarchical approach as a baseline, with the “heuristics” they describe presented as benefits over this unstructured hierarchy. Furthermore, many pure HRL papers present results in a navigation setting.
- “We show that the multi-modal memory is essential for the agent to produce good action sequences.” Based on the ablations, the multi-modal memory is “beneficial” but not “essential” as the performance differences are somewhat small.

Other notes:
- The ablations should be moved into the main body of the paper though as they are quite important and they should include variance for each approach to really understand the significance of the choices. It would also be interesting to include a human baseline for navigation to put performance into context.
- The authors should ablate for unheard sounds. I expect the audio memory, which is purely based on intensity may perform well here.


_____

Post author rebuttal:

I appreciate the author’s response and overall the authors have addressed my concerns. I am thus raising my score.

The only point that I believe still stands is #7, though I should have updated earlier. My issue with claiming this as the first use of end-to-end learned subgoals in navigation is that there have been many recent works from goal-conditioned hierarchical RL that use end-to-end learned subgoals, e.g.,
https://arxiv.org/pdf/1712.00948.pdf, https://arxiv.org/pdf/1805.08296.pdf, https://arxiv.org/pdf/1909.10618.pdf. Navigation to a known goal is a version of this problem and indeed in these works, the approaches are shown navigating between states. Others have applied end-to-end to navigation and manipulation, http://proceedings.mlr.press/v100/li20a/li20a.pdf. Overall, application of end-to-end HRL to the navigation problem is an interesting area to study, but to claim it as a major contribution I believe the paper should thoroughly examine the tradeoffs as applied to that problem, which I believe requires a detailed and standalone work.

---

> ### Author Response · Authors · 2020-11-20
> **Response to reviewer 1**
>
> We appreciate your helpful feedback.
>
> 1: Fig. 2 slightly unclear.
> Thank you.  We intended to show the graph maintained by our planner module that is constructed based on the map as the agent moves (Section 3.4). However, to avoid possible confusion we updated Fig. 2.
>
> 2: Not clear if model uses RGB or depth image.
> As shown in Fig. 2, we use depth images for building the geometric map (via the projection operation). We do not use RGB as input.  As noted in the first paragraph in Section 3.2, we use depth because it is more effective than RGB for building geometric maps (Chaplot et al., ICLR 2020). Fig. 1 is meant to illustrate the high-level idea in the paper.  We will make sure it is clear.
>
> 3: Is audio directionality used in the acoustic memory?
> The model does not explicitly encode audio directionality in the acoustic memory. However, the gradient in audio intensity stored in the memory can indicate to the agent the direction during navigation (i.e., the goal is usually in the direction of increasing audio intensity).
>
> 4: What happens if the waypoint is not possible with the graph search?
> If the waypoint is not reachable using graph search, the agent takes a single random action and breaks the planning loop. We have added this point to Section 3.4 in the updated version of the paper.
>
> 5: How are unexplored regions treated for the graph search?
> The unexplored regions are considered as free space during planning, following Chaplot et al., ICLR 2020. We have updated Section 3.4 of the paper with this point.
>
> 6: Impactfulness of this setting beyond audio-visual navigation.
> First, we believe that the AV-navigation task is impactful, in terms of both real-world applications and learning challenges for the  agent to translate raw audio-visual sensing into intelligent navigation actions. Second, please note that, unlike our approach, one of the compared methods (Gan et al., ICRA 2020) does indeed boil down the problem to point-goal navigation after predicting the goal location from audio, and it substantially underperforms the proposed approach (see Section 4).  This shows that isolating the audio as a final goal predictor is insufficient, and our model's joint learning from audio and visual throughout the entire navigation process to predict waypoints is important. Third, we agree that extensions to an audio-based semantic object navigation task could be interesting future work. We added a note about such applications to the Conclusion section.
>
> 7: Unclear or overstated contributions with respect to hierarchical policy learning.
> As we noted in Section 1, hierachical policies for navigation are not new (e.g., Chaplot et al., ICLR 2020; Stein et al., PMLR 2018; Bansal et al., CoRL 2019; Caley et al., IROS 2016). However, to our knowledge, learning to set useful subgoals in an end-to-end fashion for the navigation task is new. The novelty is learning audio-visual waypoints of auto-adaptive granularity to maximize performance. To our knowledge, this is a contribution that is orthogonal to the fact that we tackle audio-visual navigation, and can also be applied to other application settings (see the Conclusion section for examples). Our idea improves over manually-designed heuristic definitions of waypoints, as it allows the agent to be more or less conservative in waypoint selection as per the demands of the situation, as shown through our model's improved performance over the Frontier Waypoints and Supervised Waypoints baselines. Further, we couldn't find anywhere in the cited literature any claims about "heuristics" being better than "unstructured hierarchy". If we have still missed something, we would like to request the reviewer to point us to the specific paper.
>
> 8: Wording: multi-modal memory is  ''benefical" but not ''essential".
> Fair enough, the acoustic map is beneficial for the case of clean audio. However, in the presence of audio noise, it does become essential (without it, our SPL drops 0.7 points for 20 dB noise level, see Fig. 4b).  We think the results strongly justify its inclusion in the model.
>
> 9: i) Move ablation to the main body and report variance; ii) ablate for unheard sounds.
> We have updated the 'Ablations' paragraph in Section 4 of the paper with these changes and additional numbers. We report the standard deviation of each model with 5 test runs, each having a different random seed. The standard deviation is $\leq 0.5$, which is smaller than most of the improvement gains.

---

### Author Response · Authors · 2020-11-20
**Meta response for all reviewers**

We thank all the reviewers for their valuable feedback. Overall, the reviewers have appreciated our model design, comprehensive experimentation and strong results, detailed ablation studies and analyses of the model's behavior. They have also suggested some changes and asked for some clarifications. We address them in this rebuttal and by making minor revisions to the paper (highlighted in blue).

---

### Decision · Program_Chairs · 2021-01-07
**Final Decision**

**Decision:**

Accept (Poster)

**Comment:**

The paper considers a variant of the point-goal navigation problem in which the agent additionally receives an audio signal emitted from the goal. The proposed framework incorporates a form of acoustic memory to build a map of acoustic signals over time. This memory is used in combination with an egocentric depth map to choose waypoints that serve as intermediate subgoals for planning. The method is shown to outperform state-of-the-art baselines in two navigation domains.

The reviewers all agree that the paper is very well written and that the evaluations are thorough, showing that the proposed framework offers clear performance gains. The idea of combining acoustic memory as a form of map with an occupancy grid representation as a means of choosing intermediate goals is interesting. However, the significance of the contributions and their relevance are limited by the narrow scope of the audio-video navigation task, which seems a bit contrived. The paper also overstates the novelty of the work at times (e.g., being the first use of end-to-end learned subgoals for navigation). The author response resolves some of these concerns, but others remain.